# Analysis of mutational and proteomic heterogeneity of gastric cancer suggests an effective pipeline to monitor post-treatment tumor burden using circulating tumor DNA

Noriyuki Sasaki[1,2,3], Takeshi Iwaya[1,2], Takehiro Chiba[1], Masashi Fujita[4], Zhenlin Ju[5], Fumitaka Endo[1], Mizunori Yaegashi[1], Tsuyoshi Hachiya[6], Ryo Sugimoto[7], Tamotsu Sugai[7], Doris R. Siwak[8], Lance A. Liotta[9], Yiling Lu[8], Gordon B. Mills[8,10], Hidewaki Nakagawa[4], Satoshi S. Nishizuka[3]*

1 Department of Surgery, Iwate Medical University School of Medicine, Yahaba, Japan, 2 Molecular Therapeutics Laboratory, Department of Surgery, Iwate Medical University School of Medicine, Yahaba, Japan, 3 Division of Biomedical Research and Development, Iwate Medical University Institute of Biomedical Sciences, Yahaba, Japan, 4 Laboratory for Cancer Genomics, RIKEN Center for Integrative Medical Science, Yokohama, Japan, 5 Department of Bioinformatics and Computational Biology, The University of Texas, MD Anderson Cancer Center, Houston, Texas, United States of America, 6 Division of Biomedical Information Analysis, Iwate Tohoku Medical Megabank Organization, Iwate Medical University, Yahaba, Japan, 7 Department of Molecular Diagnostic Pathology, Iwate Medical University School of Medicine, Yahaba, Japan, 8 Department of Genomic Medicine, The University of Texas, MD Anderson Cancer Center, Houston, Texas, United States of America, 9 Center for Applied Proteomics and Molecular Medicine, George Mason University, Fairfax, Virginia, United States of America, 10 Department of Cell, Development & Cancer Biology, Knight Cancer Institute, Oregon Health Science University School of Medicine, Portland, Oregon, United States of America

* snishizu@iwate-med.ac.jp

**Data Availability Statement:** Underlying data are available here: Gene sequencing panel: NBDC Human Database (https://humandbs.

## Abstract

Circulating tumor DNA (ctDNA) is released from tumor cells into blood in advanced cancer patients. Although gene mutations in individual tumors can be diverse and heterogenous, ctDNA has the potential to provide comprehensive biomarker information. Here, we performed multi-region sampling (three sites) per resected specimen from 10 gastric cancer patients followed by targeted sequencing and proteomic profiling using reverse-phase protein arrays. A total of 126 non-synonymous mutations were identified from 30 samples from 10 tumors. Of these, 16 (12.7%) were present in all three regions and were designated as founder mutations. Variant allele frequencies (VAFs) of founder mutations were significantly higher than those of non-founder mutations. Phylogenetic analysis also demonstrated a good concordance between founder and truncal mutations, defined as mutations shared by all simulated clones at the trunk of the tumor phylogenetic tree. These findings led us to prioritize founder mutations for quantitative ctDNA monitoring by digital PCR with individually-designed primer/probe sets. In preoperative plasma, the average ctDNA VAF of founder mutations was significantly higher than that of non-founder mutations (p = 0.039). Proteomic heterogeneity was present across the tumor regions both within and between patients independent of mutational status. Our results suggest that, in practice, mutations having high VAF identified without multi-regional sequencing may be immediately useful for quantitative

biosciencedbc.jp/en/hum0232-v1) Submission: JGA00000000271 Study: JGAS00000000231 Dataset: JGAD00000000325 RPPA data: The University of Texas MD Anderson Cancer Center RPPA data repository URL: https://tcpaportal.org Accession ID: TCPA00000006-2.

**Funding:** SSN JP16H01578, 15KK0317, 19K09224, 19K09130, and 17K10605 Grant-in-Aid for Scientific Research URL: https://www.jsps.go.jp/english/index.html Iwate Prefectural Regional Innovation Grant https://www.pref.iwate.jp/ The funders had no role in study design, data collection and analysis, decision to publish, or preparation of the manuscript.

**Competing interests:** T. Iwaya receives research grants from Nippon Kayaku, Chugai Pharmaceutical, and Daiichi Sankyo. T. Hachiya is an employee and a Board Member of Genome Analytics Japan, Inc. G. B. Mills receives research grants from AstraZeneca, Karus Therapeutics, Nanostring, Pfizer, Tesaro, Adelson Medical Research Foundation, Breast Cancer Research Foundation, Komen Research Foundation, Ovarian Cancer Research Foundation, and Prospect Creek Foundation; serves as a consultant/scientific advisory board of AstraZeneca, Chrysalis(∗), ImmunoMET, Ionis, Lilly USA, Mills Institute for Personalized Care(∗), Nuevolution(∗), PDX Pharma, Signalchem Lifesciences, Symphogen, and Tarveda (∗, travel reimbursement only); is a stockowner of Catena Pharmaceuticals, ImmunoMet, SignalChem, Spindletop Ventures, and Tarveda; and has a relationship in licensed technology to HRD assay to Myriad Genetics, and DSP to Nanostring. S. S. Nishizuka receives research grants from Array Jet, Geninus, Taiho Pharmaceuticals, and Boelinger-Ingelheim, Geninus; serves as a consultant of CLEA Japan; receives travel expense reimbursement from Mills Institute for Personalized Care, Nomura Jimusyo; and receives honoraria from Chugai Pharmaceutical. These do not alter our adherence to PLOS ONE policies on sharing data and materials.

ctDNA monitoring but do not provide sufficient information to predict the proteomic composition of tumors.

## Introduction

Circulating tumor DNA (ctDNA) is released from tumor cells into blood [1,2]. However, the majority of ctDNA shows very low variant allele frequencies (VAFs). Therefore, a highly-sensitive method to measure ctDNA is needed for use in applications involving personalized tumor markers to predict relapse/regrowth, evaluate drug therapy efficacy, and confirm disease-free states [3–6].

Hotspot mutations in *EGFR*, *KRAS*, and *BRAF* that can provide molecular targets for drug therapies have been used for patients with lung cancer, colorectal cancer, and malignant melanoma, respectively [7–9]. Currently, peptide nucleic acid-locked nucleic acid PCR, Scorpion-amplification refractory mutation system PCR, and Cycling probe PCR are used to detect very low VAFs ($< 1\%$) for drug selection [10–12]. However, clinical applications using ctDNA are not limited to predicting utility of molecular targeting drugs; instead, the VAF dynamics of ctDNA can also be applied for monitoring of tumor burden during the course of standard, targeted or immune-therapy. Tumor burden monitoring in daily practice requires multiple time-points, rapid turnaround time, simple procedures, and economical assays. Previously, we proposed a pipeline to track tumor burden in combination with panel sequencing followed by ctDNA monitoring using digital PCR (dPCR) to identify patient-unique mutations [13–15]. However, whether the mutations identified by this approach represent the complete spectrum of tumor genetic heterogeneity that can be reflected in ctDNA monitoring is unclear.

In this study, a multi-regional panel sequence analysis of three regions in 10 gastric primary tumors was performed to clarify if ctDNA effectively represents tumor genetic heterogeneity in this disease and would allow development of highly-personalized tumor markers for tumor burden monitoring. We designed and synthesized specific primers and mut/wt probes for monitoring ctDNA using dPCR with 0.01% VAF detection limit. In addition, reverse-phase protein arrays (RPPAs) were used to examine correlations between protein level and non-synonymous single nucleotide variants (SNVs) in individual tumor specimens to determine whether the genomic heterogeneity was sufficient to predict proteomic heterogeneity. Based on integrated analysis, we developed an effective ctDNA-based tumor burden monitoring pipeline for gastric cancer patients.

## Methods

### Human samples and study design

This study was approved by the Clinical Research Ethics Committee of Iwate Medical University in compliance with the Helsinki declaration (HGH28-6). Individual written consent was obtained from each participant and all analyses were performed on deidentified samples.

Prior to the study, sample size and power calculation were not performed because no null hypothesis was set for the present observational study. The sample size restrictions were only applied for the minimum number of multiregional samples (i.e., three). The patient registration period was permitted between May 2016 and February 2018 (HGH28-6). The eligibility of participating patients included those who underwent surgical resection for $>$ cStage II gastric cancer with curative-intent at Iwate Medical University Hospital between July 2016 and January 2017, in the present observational study. The median observation period was 920 days

(range 176–1,005 days; S1 Table). No patients had previous history of any treatment at the time of informed consent. Pre- and post-operational plasma, primary tumor, and peripheral blood mononuclear cells (PBMCs) were collected from all enrolled patients.

The primary tumor specimens were taken from three regions immediately after the operation, and the cellularity of all specimens was microscopically confirmed to be ≥30% (S1 Fig, S2 Table). Each specimen was divided into two pieces for use in mutation screening and RPPA analysis and stored individually at -80˚C. DNA was extracted from primary tumors using QIAamp DNA Mini Kits (Qiagen, Hilden, Germany).

For cases suspected of having mismatch repair deficiency (dMMR) based on histopathological examinations, immunohistochemical staining using primary antibodies against MutL homolog 1 (MLH1, G168-15, BD Biosciences, Bedford, MA), MutS homolog 2 (MSH2, FE11, Calbiochem, La Jolla, CA), PMS1 homolog 2, mismatch repair system component (PMS2, C-20, Santa Cruz Biotechnology, Dallas, TX), and MutS homolog 6 (MSH6, EPR3945, Abcam, Cambridge, UK) followed by a colorimetric detection (Dako Envision system, Agilent, Santa Clara, CA) was performed.

Before and at various time points after surgery, 16 ml of blood was collected in Cell-free DNA BCT® tubes (Streck, La Vista, NB) for ctDNA monitoring. The tubes were centrifuged at 1,800 $g$ for 20 min at room temperature, and the upper phase was transferred to a 5 ml tube labeled with the patient-unique identification number for immediate storage at -80˚C until DNA isolation. Total genomic plasma DNA was extracted using the QIAamp Circulating Nucleic Acid Kit for plasma (Qiagen). At the time of preoperative blood collection, PBMCs were extracted after transfer to BD Vacutainer CPT blood collection tubes (Becton, Dickinson and Company, East Rutherford, NJ) from Cell-free DNA BCT® tubes within seven days, and the DNA was extracted using QIAamp kit for PBMCs (Qiagen). Postoperative blood samples were drawn on postoperative days 1, 7 and 30, as well as at periodic visits thereafter. The quantity of extracted DNA was measured using the Qubit® 2.0 dsDNA high sensitivity assay (Thermo Fisher Scientific, Waltham, MA).

## Target sequencing

A total of 200 ng DNA was sheared using a Covaris ultrasonicator (Woburn, MA), and the fragment size distribution was evaluated using a Bioanalyzer 2100 (Agilent Technologies, Santa Clara, CA). Sequencing libraries were prepared using ClearSeq SS Comprehensive Cancer kits (Agilent Technologies, Santa Clara, CA) according to the manufacturer's instructions. The ClearSeq Comprehensive Cancer panel targets 151 disease-associated genes that have been implicated in studies of a wide range of cancers (https://www.chem-agilent.com/pdf/ClearSeqComprehensiveCancerDataSheet_5991-5573EN.PDF). Before the sequencing run, the captured DNA library was checked for quality and quantity. The libraries were sequenced on an Illumina HiSeq 2000 platform (Illumina, Inc., San Diego, CA) according to the manufacturer's recommendations.

## Variant calling

Reads were adaptor-trimmed using Cutadapt [16] and mapped to GRCh37 using Burrows-Wheeler Aligner [17]. PCR duplicates were removed using Picard (https://broadinstitute.github.io/picard/). Low-quality reads were filtered based on mapping quality, number of mismatches and insertion and/or deletion mutations (INDELs). Improper reads were filtered based on discordance among chromosomes as well as direction and distance of paired-end reads. SNVs and INDELs were called using VarScan2 [18] with a minimum read depth of 20, a minimum VAF of five %, a minimum of four supporting reads, and a p-value threshold of

0.05. The variants were annotated using Ensembl VEP. Copy number analysis was performed using VarScan2 and DNAcopy [19].

## Copy number variation

Copy number variation (CNV) was calculated using ONCOCNV obtained via GitHub [20], with BAM files as input. Read counts in tumor BAM files were normalized, corrected for GC-content, and the CNV was detected by comparison with the baseline copy number. The baseline copy number (CN) was defined based on PBMC BAM files and was subsequently used for CNV calculation on all multiregional samples. CN segmentation was performed using the DNAcopy package of R/Bioconductor. ONCOCNV was run on the SHIROKANE supercomputer at the University of Tokyo Institute of Medical Science.

## Phylogenetic tree

Phylogenetic trees were constructed using a modified version of Canopy (version 1.3.0), an open source R package (https://cran.r-project.org/web/packages/Canopy/) [21]. A list of somatic SNV/INDELs was used as input. Based on preliminary simulations of evolutional trees, SNV data were prioritized for the simulation in the present study in which CNV data were limited from the panel sequencing. Clustering of SNV/INDELs was performed during preprocessing to accelerate simulation convergence. Markov Chain Monte Carlo simulations were run for 10 times, and the maximum simulation length was 100,000 steps. Convergence of simulations was confirmed by visually inspecting the time course of log likelihood and acceptance rate in all cases. Data in the burn-in phase were discarded.

## Digital PCR

Mutation-specific primer/probe sets were synthesized using Hypercool Primer & Probe™ technology, which is specialized to design primers/probes for short amplicons (< 70bp) by modifying bases with 2-amino dA and 5-Methyl-dC to theoretically increase the $Tm$ value (Nihon Gene Laboratories, Inc., Sendai, Japan). The PCR reaction mixture contained 7.5 μL QuantStudio™ 3D Digital PCR Master Mix, 1.5 μL 10× primer/probe mixture and 6.0 μL diluted DNA that was loaded onto a QuantStudio™ 3D Digital PCR Chip having 20,000 mini-chambers. The chips were then loaded into a ProFlex™ 2× Flat PCR System. PCR was performed using this system with 10 minutes at 96˚C, followed by 39 cycles of 60˚C for two minutes, 98˚C for 30 seconds, 30˚C for two minutes, and holding at 10˚C. The absolute count of the amplified fragment was determined using the QuantStudio 3D Digital PCR System (Thermo Fisher Scientific) and analyzed with QuantStudio 3D AnalysisSuite Cloud Software (Thermo Fisher Scientific).

## Validation of primer/probe set for digital PCR

The validation of the primer/probe sets was performed using primary tumor DNA known to have target mutations found in the panel sequence. For the founder mutation primer/probe sets, validation was performed on the sample having the highest VAF in the primary tumor. For non-founder mutation primer/probe sets, validation was performed on samples from all three regions (S4 Table).

## RPPA

Tissue samples were serially diluted two-fold (undiluted, 1:2, 1:4, 1:8, and 1:16) and arrayed on nitrocellulose-coated slides to produce sample lysate spots. Signals from the sample spots were then developed via an immunochemical reaction and tyramide-based signal amplification

before visualization with a GenPoint DAB colorimetric reaction (Agilent Technologies, Santa Clara, CA). The developed slides were scanned on a Huron TissueScope scanner to produce 16-bit TIFF images. Sample spots in the TIFF images were identified and their densities were quantified by an Array-Pro Analyzer (Meyer Instruments, Houston, TX). Relative protein levels for each sample were determined by interpolating each dilution curve produced from the densities of the 5-dilution sample spots using a "standard curve" (SuperCurve) for each slide (per antibody) [22]. A SuperCurve was constructed using an R script written by the University of Texas MD Anderson Cancer Center Department of Bioinformatics and Computational Biology (The R Foundation for Statistical Computing, Vienna, Austria). All data for relative protein levels were normalized for protein loading transformed to $log_2$ values and then median-centered for hierarchical clustering analyses. The heatmaps were developed by the University of Texas MD Anderson Cancer Center Department of Bioinformatics and Computational Biology; In Silico Solutions (Falls Church, VA); Santeon (Reston, VA); and SRA International (Arlington, VA).

### Statistical analysis

Clinicopathological and sequencing values and frequencies were analyzed using the Chi-squared test, Fisher's exact test, Student's *t*-test, Mann-Whitney U test or the Pearson's correlation coefficient, depending on the subject groups. For all statistical analyses, a two-tailed P-value of 0.05 or less was considered statistically significant. Prism 8 (GraphPad Software Inc., La Jolla, CA) was used for statistical analysis.

## Results

### Target sequencing interpretation

Between July 2016 and January 2017, ten gastric cancer patients with >cStage II disease who were to be surgically treated were enrolled (S1 Table). Target sequencing using a HiSeq 2000 System (Illumina, Inc., CA) conducted for 151 genes that have been implicated in a wide range of cancers yielded 254,606,558 sequence reads from 30 primary tumor regions as well as corresponding PBMCs. All tumor samples were confirmed to have sufficient tumor cellularity (S1 Fig, S2 Table). The average sequence depth was >500 and the mapping rate approached 100% for all libraries; the duplication rate ranged between 10 and 20%; and the off-target rate was approximately 40%. With 5% VAF set as the minimum threshold, 363 SNVs and 480 INDELs were detected as tumor-unique based on the comparison between PBMCs and primary tumors. In all, 176 non-synonymous mutations leading to amino acid substitutions were found (S2A Fig). Mutations found in all three regions analyzed for each tumor were defined as a "founder mutation", whereas a mutation found in only one or two regions was defined as a "non-founder mutation". The concept of "founder and non-founder" mutations has also been termed "public and private" for clonal and subclonal mutations in a tumor [23,24]. These terms are based on binarized data (i.e., the presence or absence of mutations), whereas the term "truncal", which is used for the results presented in the phylogenetic tree, is based on simulations using VAF data as continuous variables. Overall, 16 founder mutations, 18 non-founder mutations (two regions), and 92 non-founder mutations (one region) were identified (Fig 1). The majority of the non-founder mutations were found in two hypermutators with 92% of mutations in 8 non-hypermutators being found in all three regions. At least one founder mutation was present in 9 of the 10 registered tumors. The average VAF of founder mutations was significantly higher than that of non-founder mutations (26.5% vs 13.5%; *p* <0.0001, S2B Fig). Although some cases exhibited multiple alterations in a single gene, the total number of altered genes in this study was 77. Therefore, the average mutation per case

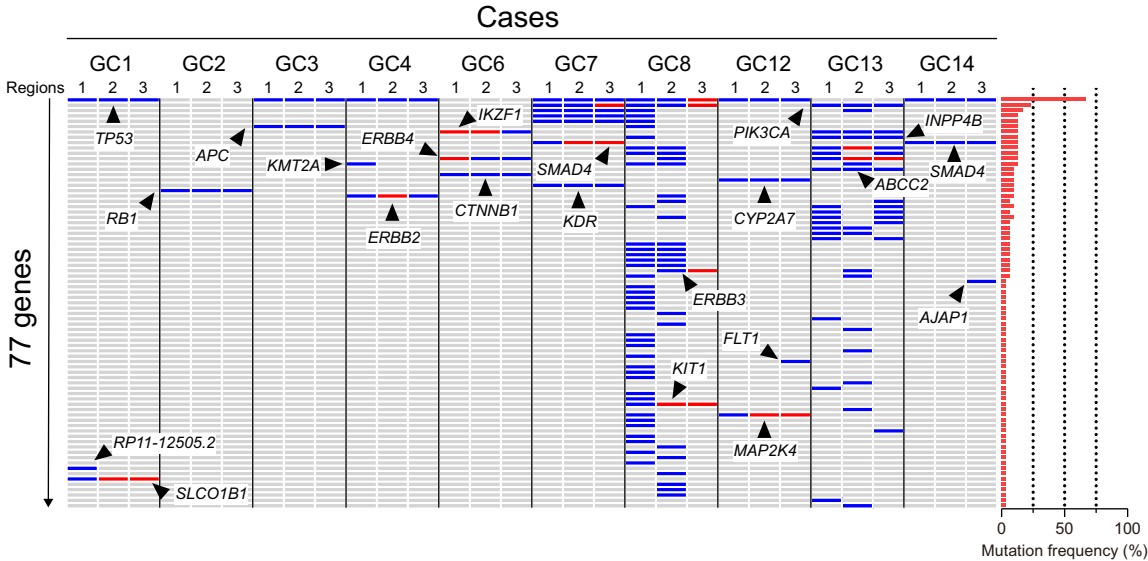

**Fig 1. Multi-region target sequencing.** Primary tumor sequencing of 151 cancer genes in samples from 10 gastric cancer patients was performed and profiles of somatic mutations in three regions are shown. Mutation frequencies of each gene in this study population are shown as red bars on the right. Mutations that were detected only by digital PCR are shown in red. Arrows indicate notable mutations. Patient GC8 and GC13 represent hypermutators. Mutations seen in more than one region in a tumor may not always represent the same mutation. The following genes/case include different mutations in at least one of three regions of a tumor: *PIK3CA*/GC13, *INPP4B*/GC13 and *ABCC2*/GC13.

was 7.7 out of the 151 genes in the sequencing panel. Frequently mutated genes across the 30 specimens included *TP53* (20/30, 67%), *PIK3CA* (7/30, 23%), and *MAP3K1* (5/30, 17%), in addition to those commonly found across all patients: *TP53* (7/10, 70%), *PIK3CA* (3/10, 30%), and *MAP3K1* (3/10, 30%).

Two cases (GC8 and GC13) had >10.2 mutations/Mb and were hypermutators according to criteria defined by Campbell et al. [25]. There was no clear association with sequencing depth or mapping rate in defining hypermutators. For these cases, immunohistochemistry for MLH1, PMS2, MSH2, and MSH6 demonstrated that at least two of these MMR proteins was absent in tumor, consistent with dMMR (S3 Fig).

## Copy number variation

Since a sequencing panel was used to identify SNVs and INDELs, the ability to assess CNVs was less comprehensive than whole genome sequencing. Based on sequencing results from the current cancer panel, the average number and size of CNVs detected was 16.6 events and 19.3 Mb, respectively. We used the ONCOCNV algorithm to identify CNVs from gene panel sequencing [20]. The CNVs of multi-region samples showed a similar trend within each patient (Fig 2A, S4 Fig). Correlation coefficients of CNVs between two arbitrary regions from a tumor was >0.6 in 70% of tumors (Fig 2B, S5 Fig). Based on the normalized CN across 30 samples, 217 and 38 genetic regions were identified as having a gain and loss of CN, respectively (S6 Fig). Overall, the high correlation among sample regions and notable CNVs suggests that CNVs, including some having potentially critical functions, may have been introduced at a relatively early stage of tumor development.

## Phylogenetic tree

Phylogenetic trees of all 10 tumors were constructed with SNVs using a modified algorithm in Canopy_1.3.0 [21]. The Canopy algorithm was used to identify 3–4 genetically different clonal

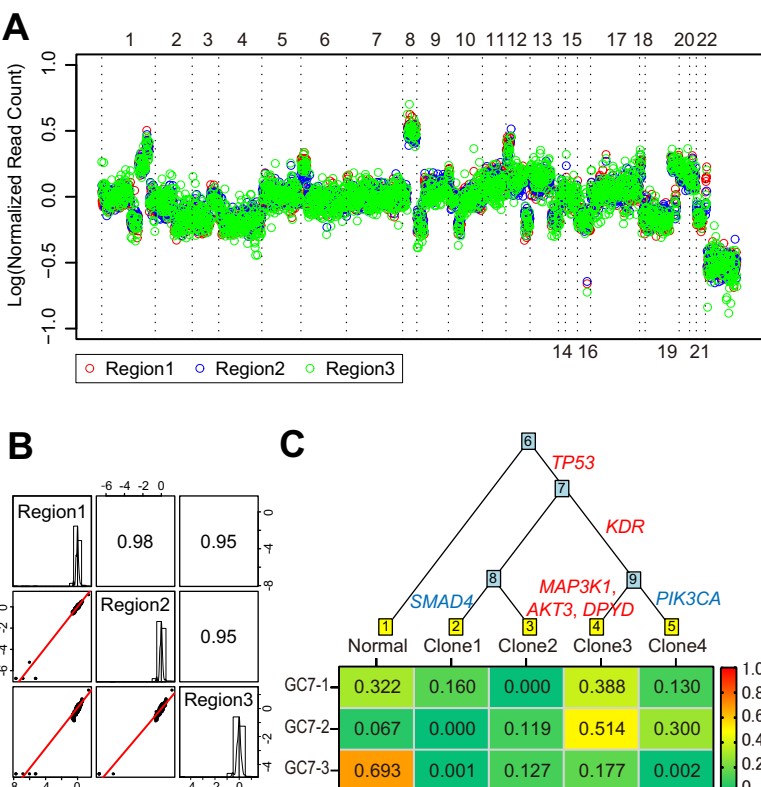

**Fig 2. Copy number alterations and a phylogenetic tree of genes in samples from gastric cancer patients.** (**A**) Alterations in GC2 copy number across all chromosomes. Horizontal and vertical axes show chromosome number and copy numbers represented by the log of normalized read count, respectively. (**B**) Pairwise scatter plots of copy number alterations between three regions of GC2. Histograms show the copy number distribution of each region. (**C**) Phylogenetic tree of GC7 based on SNV data and generated using the Canopy program. Founder and non-founder mutations are shown in red and blue, respectively. The possible clones and genomically normal cells that formed the tumor are indicated at the bottom of the tree. Simulated fractions of the clones and genomically normal cells are shown in the heatmap.

populations that are present in different proportions within a tumor. Except for GC8, which had no founder mutations, the position of founder mutations in phylogenetic trees was confirmed in all cases (Fig 2C, S7 Fig). Here, the mutations shared by all clones located at the trunk of a tumor phylogenetic tree were defined as truncal mutations [23,26]. Five of eight tumors (62.5%, except for hypermutator) had at least one truncal mutation that was a founder mutation in the tumor. Of the 16 founder mutations, 12 (75%) were located in the internal branch. These results suggest that founder mutations occur earlier in tumor development.

## Validation of primer/probe set for digital PCR

To ensure reliable signal detection for ctDNA monitoring using dPCR, mutations unique to individual tumors were selected from SNVs (including synonymous mutations) and INDELs (S3 Table) provided the mutations were either a founder mutation or a non-founder mutation with more than 10 variant coverages in a region. Validation of the set of 26 specific primers/probes was performed using primary tumor DNA with identified mutations (S4 Table). Using the uniquely-designed primer/probe sets, mutations with high VAFs (i.e., >5%) were detected in 100% (31/31) of primary tumors, whereas mutations for which the majority had low VAFs were also detected in 96% (23/24) of primary tumors; these mutations were not detected by

Next Generation Sequencer (NGS, Fig 1, S8 Fig). Among those validated with dPCR, 93% (14/15) non-founder mutations were found in all three regions. These findings suggest that there may be discrepancy between the VAFs of mutations detected using NGS and those that can be detected with dPCR. An exhaustive validation of NGS-based multiregional sequencing using highly sensitive dPCR will be warranted for establishing true founder mutations.

## Proteomic profiles of primary tumors

Levels of 293 proteins were analyzed with RPPA from the specimens neighboring those used for sequencing. Overall, the proteomic profiles were similar but not identical within the three regions assessed in each tumor. Of the nine tumors that had founder mutations, eight fell within the same protein cluster. Although GC8 had a hypermutated genomic profile without detected founder mutations within multi-region samples, overall the proteomic profiles were relatively consistent within the set of multi-region samples even in this tumor (Fig 3A, S9 Fig). GC7 had five founder mutations, suggesting that the tumor might have been genetically less heterogeneous than the other tumors. Interestingly, however, one of the multiregional samples (GC7-3) for this tumor appeared in a separate cluster. PAK1, ARID1A, 53BP1 and eIF4G had distinct levels in GC7-3 relative to the other two GC7 samples representing a potentially functional difference in the regions (S9 Fig). RPPA data have been deposited at https://tcpaportal.org (TCPA000000006-2).

## Protein level prediction by gene mutation

Previous reports have suggested that protein abundance measured by mass spectrometry cannot be reliably predicted from DNA measurements [27–29]. In the present study, from the set of 151 genes and 293 proteins, 42 gene-protein matched pairs were compared to examine whether mutant genotype affected protein levels of the matching protein. Twelve pairs of wild type and mutated genes including *TP53*-p53, *PIK3CA*-PI3K 110α, *ATM*-ATM, *ATM*-ATM_pS1981, *SMAD4*-SMAD4, *INPP4B*-INPP4b, *CTNNB1*-β-Catenin, *CTNNB1*-β-Catenin_pT41_S45, *KDR*-VEGFR, *RB1*-Rb_pS807_S811, *ERBB2*-HER2, and *ERBB2*-HER2_pY1248 were compared using a $t$-test. p53 protein levels were higher in *TP53* mutated samples, suggesting that p53 protein levels were stabilized by the *TP53* mutations ($p = 0.0004$, Fig 3B). In contrast, none of the other eleven proteins, including phosphoproteins, showed higher levels in the mutated samples than wild type (Fig 3B). Of note, a Mann-Whitney U test showed that Rb_pS807_S811 protein level was different between *RB1* mutated and wild-type samples ($p = 0.0054$).

## Founder mutation detection in ctDNA

Analysis with ctDNA of 26 mutations was performed by dPCR on 92 plasma samples taken during the clinical course of management of the 10 patients. The median DNA concentration was 11.2 (5.0–29.7) ng/ml in preoperative plasma. Specific mutations consistent with the primary tumor were detected in preoperative plasma for 3/10 cases (eight mutations overall, average VAF 0.73%, S4 Table). The average ctDNA VAF of founder mutations was higher than that for non-founder mutations (1.09% vs 0.29%; $p = 0.0039$, S10A Fig). Quantified pretreatment ctDNA VAF by dPCR strongly correlated with VAF of the primary tumor as quantitated by NGS ($r = 0.9136$, $p = 0.0006$, S10B Fig). There were no significant differences in age or tumor size between the ctDNA-negative and -positive groups (S10C Fig).

## Longitudinal ctDNA monitoring

GC1 (Fig 4A) underwent gastrectomy with curative-intent and received no post-operative adjuvant chemotherapy. In preoperative ctDNA, *TP53* VAF was 1.33%. The VAF of *TP53*

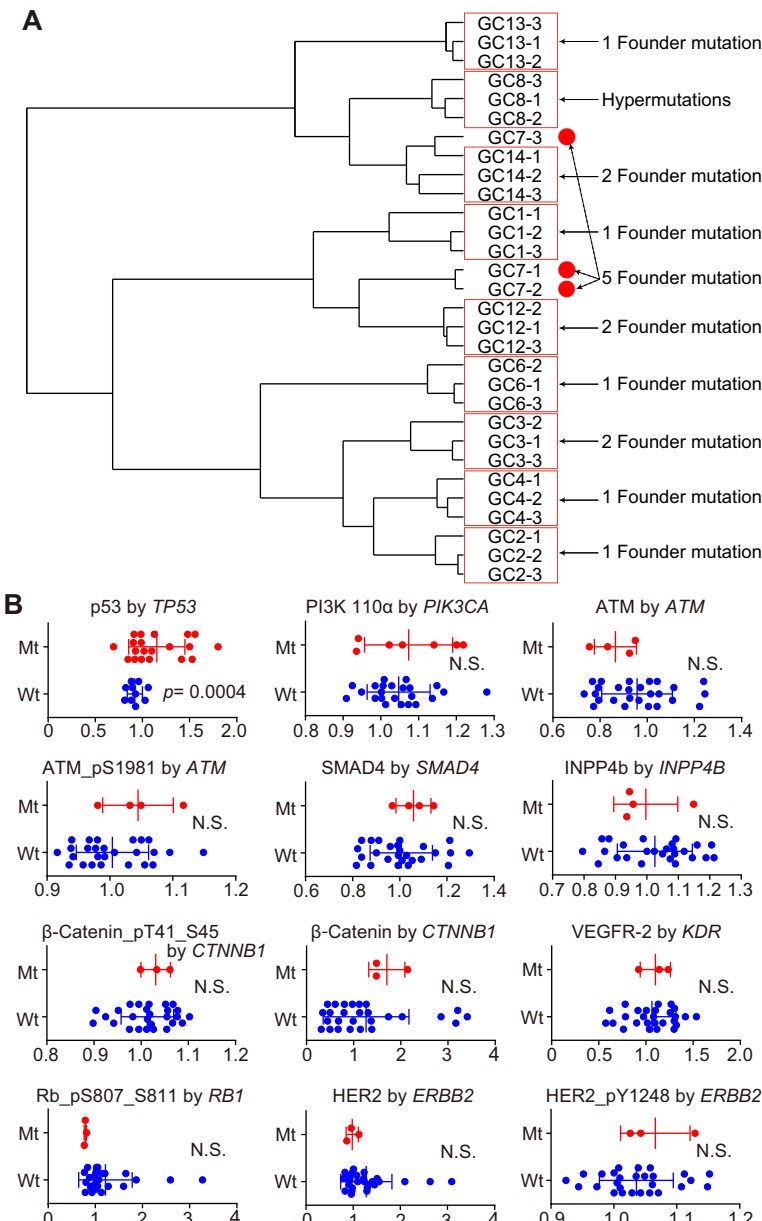

**Fig 3. Protein levels based on gene mutation status.** (**A**) Dendrogram for sample clustering. Red boxes denote multiregional samples from a tumor that fall into the same cluster. The number of founder mutations for each tumor are shown on the right. (**B**) Comparison of protein levels based on mutational status of coding genes. The horizontal axis shows arbitrary units from RPPA and the vertical axis shows mean values and values ± 2SD from the mean. The combinations include: *TP53*-p53, *PIK3CA*-PI3K 110α, *ATM*-ATM, *ATM*-ATM_pS1981, *SMAD4*-SMAD4, *INPP4B*-INPP4b, *CTNNB1*-β-Catenin, *CTNNB1*-β-Catenin_pT41_S45, *KDR*-VEGFR, *RB1*-Rb_pS807_S811, *ERBB2*-HER2, *ERBB2*-HER2_pY1248. The *p* values were calculated by Student's *t*-test. Wt, wild type. Mt, mutant type. NS, Not Significant.

ctDNA dropped below the detection limit immediately after surgery and remained unchanged for at least 919 days.

GC12 (Fig 4B) underwent total gastrectomy with curative-intent followed by capecitabine/oxaliplatin adjuvant chemotherapy, which was discontinued due to adverse events. *LAMA2* and *MAP2K4* ctDNA that were detectable prior to surgery became undetectable after surgery.

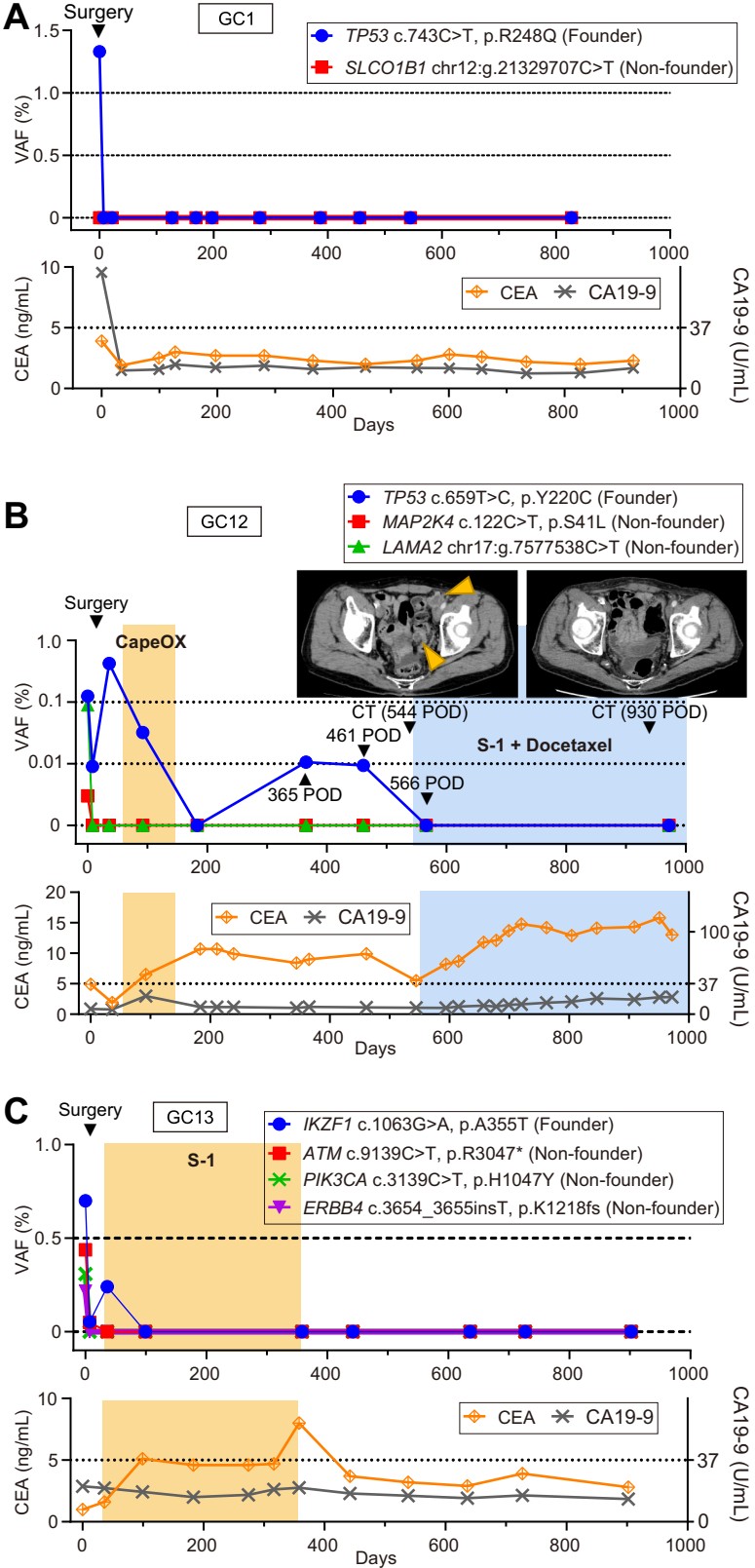

**Fig 4. Clinical courses with ctDNA monitoring.** VAFs of ctDNA are indicated in comparison with serum tumor markers, treatment, and tumor images by CT, if available. The chemotherapeutic term is indicated by colored boxes.

(**A**) The ctDNA fraction measured for both *TP53* (c.743C>T, p.R248Q) and *SLCOB1* (C>T, chr12:21329707). (**B**) The ctDNA fraction measured for *TP53* (c.659T>C, p.Y220CT), *MAP2K4* (c.122C>T, p.S41L) and *LAMA2* (C>T). Computed tomography images display the status of peritoneal disseminations, indicated by arrowheads. (**C**) The ctDNA fraction measured for *IKZF1* (c.1063G>A, p.A355T), *ATM* (c.9139C>T, p.R3047*), *PIK3CA* (c.3139C>T, p. H1047Y) and *ERBB4* (c.3654_3655insT, p.K1218fs). CapeOX: capecitabine + oxaliplatin. Capecitabine, an oral fluoropyrimidine. Oxaliplatin, a third-generation platinum complex. S-1, an oral fluoropyrimidine. VAF, variant allele frequency. CEA, carcinoembryonic antigen. CA19-9, carbohydrate antigen 19–9.

The postoperative CEA level remained >5.0 ng/ml (i.e., positive) for an extended period of time after surgery. Although samples were positive for *TP53* ctDNA at 365 days and 461 days after surgery, no sign of recurrence was confirmed by computed tomography (CT) examinations. The CT at 544 days after surgery diagnosed a recurrence in the peritoneum as evidenced by the presence of nodular lesions in the pelvic cavity. Immediately after the diagnosis of this recurrence, S-1/Docetaxel chemotherapy was initiated. The VAF of ctDNA at 566 days after surgery dropped to undetectable levels. At 972 days after surgery, no obvious nodular lesions were confirmed by CT and ctDNA remained undetectable.

GC13 (Fig 4C) underwent total gastrectomy with curative intent followed by adjuvant S-1 monotherapy for one year. *IKZF1*, *ATM*, *PIK3CA*, and *ERBB4* ctDNA were detected before the operation, but the levels of these genes were undetectable after the operation and remained so for up to 902 days. No signs of recurrence have been seen at up to 1,069 days.

The ctDNA level of the three abovementioned cases was measured with both founder and non-founder mutations. In general, founder mutations more precisely reflect tumor burden, as seen by the steep drop after surgery. In fact, only the *TP53* mutation (i.e., the founder) exhibited a steep drop after surgery for GC1, while *SLCOB1* (i.e., non-founder) remained negative, suggesting that genetic tumor heterogeneity may have affected ctDNA release from the primary tumor. Mutations used for ctDNA monitoring for GC12 and GC13 exhibited synchronized dynamics, although the founder mutations showed the most visible contrast. The other seven cases in which ctDNA was not detected in the preoperative plasma showed no detectable ctDNA in the postoperative observation period ranging 176 to 1,005 days (S11 Fig). Six of the seven cases had highly advanced disease except for one stage IA and were monitored with ctDNA for at least one of the founder mutations. There are potential mechanical difficulties in the detection of ctDNA from cancer cells in ascites [30], but the pathological nature of gastric cancer, including the high stromal cell content and tissue background comprising an overwhelming population of genetically-normal inflammatory cells, could also contribute to the ~30% detection rate for ctDNA in gastric cancer [31–33].

## Discussion

In 30 samples from 10 gastric tumors, 126 non-synonymous mutations were identified by panel sequencing with the locations and types of mutations being diverse. Of the 10 tumors assessed, nine had at least one founder mutation defined as mutations found in three physically separate tumor regions. Our results suggest that mutations having a high VAF are likely to be founder mutations. Such founder mutations are also likely to be truncal mutations in phylogenetic terms and thus a high VAF mutation from a panel sequencing of a single tumor region or a biopsy could be a surrogate index for founder and possibly truncal mutations. Founder mutations are more likely to be detected in ctDNA, thereby mutations with a high VAF should be chosen for ctDNA monitoring. Notably, however, almost all (93%) "non-founder" mutations defined by NGS were detected in all sample regions by dPCR. Phylogenetic analysis also suggested that most of the sample regions contained all clones for which the fraction was less than 0.1%. Therefore, the definition of "founder mutation" may limited when

assessed by NGS data, whereas the true mutational heterogeneity may be ultimately quantifiable using methods having a detection limit of less than 0.1% VAF, such as dPCR [34,35] or single cell sequencing [36]. The exhaustive assessment of multiregional sequencing may alter the initial condition of the phylogenetic simulation.

In the present study, seven of the 10 cases were ctDNA-negative at any time-point. Of these seven cases, six (86%) were alive without recurrence through the latest observation. The remaining three cases were pre-operative ctDNA-positive. In two of these cases (67%), post-operative ctDNA was not detected and remained undetectable throughout the follow-up period that ranged between 902 and 919 days. In general, ctDNA monitoring in the post-treatment setting is expected to provide information about need for additional treatment as well as potential therapeutic options. However, it should be noted that *peritonitis carcinomatosa*, a highly advanced form of gastric cancer, may be an exception as was observed for GC2 and GC12. *Peritonitis carcinomatosa* may often involve either solid masses, dissemination, or ascites with a high density of cancer cells. A recent study showed that only 38.8% of overall "solid" *peritonitis carcinomatosa* patients exhibited detectable preoperative ctDNA with >0.1% VAF by NGS [37]. Among the cases enrolled in this study, GC12 had recurrent disease with measurable peritoneal nodules. The recurrence of GC2 was apparent from ascites (i.e., liquid) arising from *peritonitis carcinomatosa* and showed no detectable levels of ctDNA despite having a disease status that was as advanced as that seen for GC12. Taken together, these results suggest that solid lesions in the peritoneum may produce detectable amounts of ctDNA that are in a range similar to that seen for primary gastric tumors, even though the ctDNA level may not always reflect the degree of disease progression, particularly when the disease is disseminated. In fact, ctDNA levels of patients with ascites were reported to fall outside the general concordance of tumor volume and ctDNA level [30]. It seems that not all gastric cancer cases can be monitored by ctDNA alone. Further observational studies are warranted to determine the clinical validity of ctDNA in gastric cancer diagnosis, in comparison with current modalities, such as serum markers and imaging studies.

One of the expectations for ctDNA is to provide a rationale for selection of molecular targeting therapy based on gene mutations. If this rationale is valid, patients could receive information about drug selection based on a simple blood test. However, most drug targets are proteins and the direct effect of mutations in genes encoding these proteins is largely unknown. Here the panel sequence and RPPA analysis showed that p53 protein levels were higher in the *TP53* mutated samples presumably due to protein stabilization [38–40]. Unlike p53, for most proteins the manner in which functions are affected by gene mutations is poorly understood. Although we did not test the protein level with antibodies against specific mutant proteins, our present results generally suggest that predictions about molecular target level should not be based solely on gene mutations [41,42].

There are several limitations to this study. First, the number of gastric cancer patients was limited to 10. Second, the panel used in the present study lacked several genes that are frequently mutated in gastric cancer such as *RHOA* and *ARID1A* [43,44]. The panel sequence selection may thus have led to an underestimate of the number of founder or truncal mutations. Third, the "long tail" distribution of mutated genes restricts opportunities to estimate protein levels based on the target mutation status [45–47]. Although the multi-regional comparison indeed increased the comparable pairs, population- and efficacy-based investigations are still needed. Finally, the majority of antibody epitopes used in RPPA have not been identified. If we had used antibodies specific for mutated proteins, mutation-specific effects and functions based on the gene mutation status might have been observed at the protein level.

In summary, we demonstrated that ctDNA detection could be performed using highly sensitive dPCR with the identification of high-VAF tumor-unique mutations in gastric cancer

patients. In addition, careful consideration should be given to whether protein levels can be predicted based on gene mutations alone.

## Supporting information

**S1 Fig. Multi-region sampling.** Primary tumor tissue was taken from three regions (yellow circles) immediately after surgery, and cellularity of at least 30% was microscopically confirmed.
(SVG)

**S2 Fig. Mutation characteristics.** (**A**) Tumor-unique mutation profile according to mutation type. Colored boxes in each column denote tumor-unique mutations of an individual tumor sample. Each row represents cancer-associated genes from the ClearSeq SS Comprehensive Cancer Panel. An asterisk (*) indicates that the mutation was confirmed by dPCR but not by NGS. Mutations seen in more than one region in a tumor may not always represent the same mutation. The following genes/case include different mutations in at least one of three regions of a tumor: *PIK3CA*/GC13, *INPP4B*/GC13, and *ABCC2*/GC13. (**B**) Comparison of the mean variant allele frequency (VAF) between founder and non-founder mutations assessed by Student's *t*-test.
(EPS)

**S3 Fig. Immunohistochemistry of hypermutator (GC8 and GC13).** (**A**) GC8. (**B**) GC13. MSH2 and MSH6 protein expression was detected in the tumor cells in both cases. All scale bars indicate 200 μm.
(SVG)

**S4 Fig. Copy number variations of the log$_2$ ratio across chromosomes.** The horizontal axis shows the chromosomal number and the vertical axis shows the log$_2$ of the normalized read count. Red, blue, and green circles represent region1, region2, and region3, respectively.
(SVG)

**S5 Fig. Pairwise scatter plots of copy number alterations between three regions.** Histograms show the copy number distribution of each region.
(SVG)

**S6 Fig. Copy number variations (CNVs) detected in each sample.** Red and blue represent CN gain and loss, respectively.
(EPS)

**S7 Fig. Simulation of phylogenetic trees at the level of nonsynonymous mutations using the Canopy program.** Founder and non-founder mutations are shown in red and blue text, respectively. The possible clones and genomically normal cells forming the tumor are indicated at the bottom of the tree. Simulated fractions of the clones and genomically normal cells are shown in a heatmap below each tree.
(EPS)

**S8 Fig. Validation of target sequencing results by dPCR.** The horizontal axis indicates tumor VAFs from the ClearSeq SS Comprehensive Cancer Panel$^{®}$. Detected mutations were validated by dPCR with the corresponding mutant and wild-type probe set (vertical axis). Pearson's correlation coefficient and the corresponding *p* value are indicated. VAF, variant allele frequency.
(EPS)

**S9 Fig. Heatmap of unsupervised hierarchical clustering (unsupervised for both antibodies and samples) by RPPA.** Proteomic profiles were generally similar within a tumor. The heatmap was generated using Cluster 3.0 (http://bonsai.hgc.jp/~mdehoon/software/cluster/software.htm) and a centered metric with Pearson's correlation coefficient as a "distance" and an average-linkage method for hierarchical clustering. The resulting heatmaps were visualized in Java Treeview (http://jtreeview.sourceforge.net) and presented as high-resolution bmp files. The horizontal axis represents proteins and the vertical axis represents 30 samples with three multi-regional samples per tumor. A dendrogram for sample clustering is enlarged at the bottom. Vertical red lines adjacent to sample names indicate that the samples are in the same cluster. Representative high level proteins for major sample clusters are indicated by respective boxes.
(EPS)

**S10 Fig. Evaluation of preoperative ctDNA.** (**A**) Comparison of mean VAFs of preoperative ctDNA between founder and non-founder mutations by Student's *t*-test. (**B**) Correlation of VAFs between primary tumors by NGS and ctDNA by dPCR. The horizontal axis shows VAFs of primary tumors by NGS and vertical axes show corresponding VAFs of ctDNA by dPCR. Correlation coefficient and corresponding *p* value are indicated. (**C**) Comparisons of the mean age and tumor size between ctDNA-negative and -positive groups as assessed by Student's *t*-test. VAF, variant allele frequency.
(EPS)

**S11 Fig. ctDNA monitoring of seven cases in which preoperative ctDNA was not detected.** Values of ctDNA during the clinical course are indicated in comparison with serum tumor markers, treatment, and tumor burden, if any. Chemotherapeutic term is indicated by colored boxes. A computed tomography image displaying the status of peritoneal dissemination indicated by arrowheads is shown.
(EPS)

**S1 Table. Patient characteristics.**
(DOCX)

**S2 Table. Cellularity of multi-region samples.**
(DOCX)

**S3 Table. Tumor-unique mutations present in 10 cases.**
(DOCX)

**S4 Table. Digital PCR analysis using mutation-specific primer/probe sets.**
(DOCX)

## Acknowledgments

The authors thank all the patients for their participation in this study, as well as hospital and administrative staff. We also thank Drs. Keisuke Koeda, Yuji Akiyama, Shigeaki Baba, Yutaka Nishinari and Haruka Nikai for their assistance with patient enrollment and Drs. Yuchao Jiang and Nancy Zhang for valuable discussions regarding the Canopy algorithm.

## Author Contributions

**Conceptualization:** Hidewaki Nakagawa, Satoshi S. Nishizuka.

**Data curation:** Noriyuki Sasaki, Takeshi Iwaya, Ryo Sugimoto, Tamotsu Sugai, Lance A. Liotta, Satoshi S. Nishizuka.

**Formal analysis:** Noriyuki Sasaki, Takeshi Iwaya, Takehiro Chiba, Masashi Fujita, Zhenlin Ju, Tsuyoshi Hachiya, Doris R. Siwak, Yiling Lu, Hidewaki Nakagawa, Satoshi S. Nishizuka.

**Funding acquisition:** Satoshi S. Nishizuka.

**Investigation:** Noriyuki Sasaki.

**Methodology:** Noriyuki Sasaki, Takeshi Iwaya, Takehiro Chiba, Fumitaka Endo, Mizunori Yaegashi, Doris R. Siwak, Yiling Lu, Gordon B. Mills, Satoshi S. Nishizuka.

**Project administration:** Satoshi S. Nishizuka.

**Supervision:** Satoshi S. Nishizuka.

**Validation:** Noriyuki Sasaki, Takeshi Iwaya.

**Visualization:** Noriyuki Sasaki.

**Writing – original draft:** Noriyuki Sasaki, Satoshi S. Nishizuka.

**Writing – review & editing:** Gordon B. Mills, Satoshi S. Nishizuka.

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
