## [Decision Letter · Decision Letter 0]

13 Jul 2020

PONE-D-20-11346

Analysis of mutational and proteomic heterogeneity of gastric cancer suggests an effective pipeline to monitor post-treatment tumor burden using circulating tumor DNA

PLOS ONE

Dear Dr. Nishizuka,

Thank you for submitting your manuscript to PLOS ONE. After careful consideration, we feel that it has merit but does not fully meet PLOS ONE’s publication criteria as it currently stands. Therefore, we invite you to submit a revised version of the manuscript that addresses the points raised during the review process.

Based on external reviews and internal evaluation it was felt that the manuscript needs major revision. Please carefully address all the points raised yb the reviewers.

We look forward to receiving your revised manuscript.

Kind regards,

Obul Reddy Bandapalli, MSc, PhD

Academic Editor

PLOS ONE

Journal Requirements:

2. Please provide additional details regarding participant consent. In the ethics statement in the Methods and online submission information, please ensure that you have specified whether consent was informed.

3. In your Methods section, please provide additional information about the participant recruitment method and the demographic details of your participants. Please ensure you have provided sufficient details to replicate the analyses such as: a) a description of any inclusion/exclusion criteria that were applied to participant recruitment, b) a table of relevant demographic details and c) a statement as to whether your sample can be considered representative of a larger population.

4. Please provide a sample size and power calculation in the Methods, or discuss the reasons for not performing one before study initiation.

5. To comply with PLOS ONE submission guidelines, in your Methods section, please provide additional information regarding your statistical analyses. For more information on PLOS ONE's expectations for statistical reporting, please see https://journals.plos.org/plosone/s/submission-guidelines.#loc-statistical-reporting.

6. Thank you for stating the following in the Competing Interests section:

"I have read the journal's policy and the authors of this manuscript have the following competing interests: T. Iwaya receives research grants from Nippon Kayaku, Chugai Pharmaceutical, and Daiichi Sankyo. T. Hachiya is an employee and a Board Member of Genome Analytics Japan, Inc. G. B. Mills receives research grants from AstraZeneca, Karus Therapeutics, Nanostring, Pfizer, Tesaro, Adelson Medical Research Foundation, Breast Cancer Research Foundation, Komen Research Foundation, Ovarian Cancer Research Foundation, and Prospect Creek Foundation; serves as a consultant/scientific advisory board of AstraZeneca, Chrysalis(*), ImmunoMET, Ionis, Lilly USA, Mills Institute for Personalized Care(*), Nuevolution(*), PDX Pharma, Signalchem Lifesciences, Symphogen, and Tarveda (*, travel reimbursement only); is a stockowner of Catena Pharmaceuticals, ImmunoMet, SignalChem, Spindletop Ventures, and Tarveda; and has a relationship in licensed technology to HRD assay to Myriad Genetics, and DSP to Nanostring. S. S. Nishizuka receives research grants from Array Jet, Geninus, Taiho Pharmaceuticals, and Boelinger-Ingelheim, Geninus; serves as a consultant of CLEA Japan; receives travel expense reimbursement from Mills Institute for Personalized Care, Nomura Jimusyo; and receives honoraria from Chugai Pharmaceutical."

7. In your Data Availability statement, you have not specified where the minimal data set underlying the results described in your manuscript can be found. PLOS defines a study's minimal data set as the underlying data used to reach the conclusions drawn in the manuscript and any additional data required to replicate the reported study findings in their entirety. All PLOS journals require that the minimal data set be made fully available. For more information about our data policy, please see http://journals.plos.org/plosone/s/data-availability.

Reviewers' comments:

Reviewer's Responses to Questions

**Comments to the Author**

1. Is the manuscript technically sound, and do the data support the conclusions?

Reviewer #1: Yes

Reviewer #2: Yes

2. Has the statistical analysis been performed appropriately and rigorously? 

Reviewer #1: Yes

Reviewer #2: Yes

3. Have the authors made all data underlying the findings in their manuscript fully available?

Reviewer #1: Yes

Reviewer #2: Yes

4. Is the manuscript presented in an intelligible fashion and written in standard English?

Reviewer #1: No

Reviewer #2: Yes

5. Review Comments to the Author

Reviewer #1: The study is multifaceted so the overarching theme is unclear at times. The investigators analyzed 10 gastric cancers by performing targeted sequencing and proteomic analysis of three regions of each tumor. They identified several tumor specific mutations and protein profiles. Mutations were then analyzed in circulating tumor DNA to determine whether they could be used as biomarkers for treatment response. The data were strong and mostly support the conclusions that were drawn. However, the authors need to address several concerns.

1) The Abstract is very difficult to follow as compared to the Author Summary which was well written. I suggest that the Abstract be written so the reader can more easily follow the logic of the study.

2) The term founder and non-founder mutations is used to describe whether a mutation is present in all or some of the regions of a tumor. This terminology is fine. However, others have used the terminology of public versus private. Public is present in all regions, whereas private is present in only some.

3) The authors define non-founding as mutations based on Next Gen as being present in only SOME regions. Yet, digital PCR revealed that many non-founding mutations were actually present in all regions. Would it be better to reclassify such "non-founding" as actually "founding"? Should you use three classes - public, consistent VAF; public, low or inconsistent VAF, and private.

From the Abstract, "93% of non-founder mutations were found in all three regions". Aren't these non-founders actually founders by you definition.

4) The authors did a good job of presenting the data. Presumably the phylogenetic trees force that notion that each tumor is derived from a single rogue cell. Is there any evidence that some of these cancers could actually be derived from multiple different rogue cells? Figure 1 demonstrates that some genes are mutant in all three regions of a tumor but is the mutation always the same mutation. Please state explicitly in legend.

5) Is it apparent that mutation profile would correlate with protein heterogeneity. can other studies in which this correlation is tested be cited. It seems with targeted sequencing and limited protein analysis that finding any correlation would be quite a long shot unless very, very significant.

5) Does the ctDNA markers perform better than current markers in assessing treat response and disease free survival? Please provide a statistical analysis.

Reviewer #2: Good report, will likely significantly advance the use of circulating tumor DNA. Authors are to be commended for careful IRB record notations and excellent supporting online material file, data transparency records.

6. PLOS authors have the option to publish the peer review history of their article (what does this mean?). If published, this will include your full peer review and any attached files.

Reviewer #1: No

Reviewer #2: No

---

## [Author Response · Author response to Decision Letter 0]

31 Aug 2020

Reviewer #1:

The study is multifaceted so the overarching theme is unclear at times. The investigators analyzed 10 gastric cancers by performing targeted sequencing and proteomic analysis of three regions of each tumor. They identified several tumor specific mutations and protein profiles. Mutations were then analyzed in circulating tumor DNA to determine whether they could be used as biomarkers for treatment response. The data were strong and mostly support the conclusions that were drawn. However, the authors need to address several concerns.

>Thank you very much for your constructive comments. In particular, we are encouraged by the words, “The data were strong and mostly support the conclusions that were drawn”. We believe that we have now fully responded to the points raised.

1) The Abstract is very difficult to follow as compared to the Author Summary which was well written. I suggest that the Abstract be written so the reader can more easily follow the logic of the study.

>The Abstract has been largely rewritten based on the Author Summary. Thank you for your suggestion.

Circulating tumor DNA (ctDNA) is released from tumor cells into blood in advanced cancer patients. Although gene mutations in individual tumors can be diverse and heterogenous, ctDNA has the potential to provide comprehensive biomarker information. Here, we performed multi-region sampling (three sites) per resected specimen from 10 gastric cancer patients followed by targeted sequencing and proteomic profiling using reverse-phase protein arrays. A total of 126 non-synonymous mutations were identified from 30 samples from 10 tumors. Of these, 16 (12.7%) were present in all three regions and were designated as founder mutations. Variant allele frequencies (VAFs) of founder mutations were significantly higher than those of non-founder mutations. Phylogenetic analysis also demonstrated a good concordance between founder and truncal mutations, defined as mutations shared by all simulated clones at the trunk of the tumor phylogenetic tree. These findings led us to prioritize founder mutations for quantitative ctDNA monitoring by digital PCR with individually-designed primer/probe sets. In preoperative plasma, the average ctDNA VAF of founder mutations was significantly higher than that of non-founder mutations (p = 0.039). Proteomic heterogeneity was present across the tumor regions both within and between patients independent of mutational status. Our results suggest that, in practice, mutations having high VAF identified without multi-regional sequencing may be immediately useful for quantitative ctDNA monitoring but do not provide sufficient information to predict the proteomic composition of tumors.

2) The term founder and non-founder mutations is used to describe whether a mutation is present in all or some of the regions of a tumor. This terminology is fine. However, others have used the terminology of public versus private. Public is present in all regions, whereas private is present in only some.

>Thank you for pointing this out. In fact, we understand the terminology of public versus private, which are fully matched conceptually to what we had intended to define founder and non-founder, respectively (Gerlinger et al, NEJM, 2012; Sottoriva et al, Nat Genet, 2015). The definition of founder mutations has also been termed clonal, ubiquitous, and shared (Trajlic et al, BBA, 2015; Saito et al, Nat Commun, 2018). However, of note, these terms are based on binarized data (i.e., the presence or absence of mutations) whereas the term “truncal”, which is the term used for the results presented in the phylogenetic tree, is based on simulations using VAF data as continuous variables. Therefore, we differentiated between the concept of founder, public, clonal, ubiquitous, and shared mutations from truncal mutations. We have added a list of references to describe equivalent observations as our definition for founder and non-founder mutations. Please see Page 12, Line 229.

“The concept of “founder and non-founder” mutations has also been termed “public and private” for clonal and subclonal mutations in a tumor　[23, 24]. These terms are based on binarized data (i.e., the presence or absence of mutations), whereas the term “truncal”, which is used for the results presented in the phylogenetic tree, is based on simulations using VAF data as continuous variables.”

3) The authors define non-founding as mutations based on Next Gen as being present in only SOME regions. Yet, digital PCR revealed that many non-founding mutations were actually present in all regions. Would it be better to reclassify such "non-founding" as actually "founding"? Should you use three classes - public, consistent VAF; public, low or inconsistent VAF, and private.

From the Abstract, "93% of non-founder mutations were found in all three regions". Aren't these non-founders actually founders by you definition.

>Thank you very much for raising this important point. We validated only 15 of the 110 non-founder mutations for which dPCR primer/probe sets were designed. Therefore, we have not been able to fully categorize our NGS data as you suggested at present. We agree with your point and indeed had an impression that the “true” non-founder mutations may be fewer than we have seen if we use a highly sensitive method, such as dPCR. The sentence from the original abstract “93% of non-founder mutations were found in all three regions” can only be applied for “Among those validated” samples (n=15). Hence, it may not be generalized with the present data. We have deleted the sentence in the updated abstract. The following sentence has been added on Page 16, Line 308.

“An exhaustive validation of NGS-based multiregional sequencing using highly sensitive dPCR will be warranted for establishing true founder mutations.”

4)-1 The authors did a good job of presenting the data. Presumably the phylogenetic trees force that notion that each tumor is derived from a single rogue cell. 

>Thank you very much for your nice compliment. As you presumed, the pre-simulation assumption for phylogenetic trees coded by a modified Canopy algorithm is that all cancer clones ultimately originate from a normal cell. In addition to the absence of whole genome sequencing for all regional samples, we must admit that there is a limit of phylogenetic tree simulation due to the nature of the Canopy algorithm stated in the original article, “we do not directly observe the clones; instead the samples we sequence are mixtures” (Jiang et al, PNAS, 2016). A relevant sentence was inserted in the Discussion section on Page 21, Line 430.

“The exhaustive assessment of multiregional sequencing may alter the initial condition of the phylogenetic simulation.”

4)-2 Is there any evidence that some of these cancers could actually be derived from multiple different rogue cells?

In response to the interesting point that the reviewer raised, in fact, we observed some tumors that do not have a truncal mutation, including GC2, GC6, GC8 (hypermutator), GC13 (hypermutator), and GC14, while most of them had founder mutations. These observations may be because the tumor was derived from multiple different transformed cells. However, the observations were not exhaustive to fully conclude that there may be multiple origins of a tumor. Rather, it may be likely due to technical limitations from the definition of founder (based on binary variables) and truncal (based on continuous variables) mutations.

4)-3 Figure 1 demonstrates that some genes are mutant in all three regions of a tumor but is the mutation always the same mutation. Please state explicitly in legend.

>Thank you for the suggestion. Three genes include different mutations in at least one of three regions of a tumor: PIK3CA, INPP4B, and ABCC2 of GC13. INPP4B and ABCC2 were not marked with arrows as notable mutations in Figure 1 in the original version. Primer/probe sets for digital PCR were not designed for these two mutations. We added the following sentence to the Figure 1 and Figure S2 legends as well as arrows indicating INPP4B and ABCC2 of GC13 in Figure 1. However, please note that the GC13 is considered to be a hypermutator, which may not be representative of tumor clonal evolution.

“Mutations seen in more than one region in a tumor may not always represent the same mutation. The following genes/case include different mutations in at least one of three regions of a tumor: PIK3CA/GC13, INPP4B/GC13, and ABCC2/GC13.”

5)-1 Is it apparent that mutation profile would correlate with protein heterogeneity. can other studies in which this correlation is tested be cited. 

>Thank you for pointing this out. We unintentionally missed the necessary citations. We have now cited previous proteogenomic studies that have suggested that protein abundance cannot be reliably predicted from DNA-level measurements (Zhang et al, Nature, 2014; Mertins et al, Nature, 2016; Mun et al, Cancer Cell, 2018). We have inserted a relevant paragraph and citations in the Results subsection, “Protein level prediction by gene mutation” on Page 17, Line 338.

“Previous reports have suggested that protein abundance measured by mass spectrometry cannot be reliably predicted from DNA measurements [27-29]. In the present study, ……..”

5)-2 It seems with targeted sequencing and limited protein analysis that finding any correlation would be quite a long shot unless very, very significant.

>We agree with the reviewer’s comment. It is a long shot. Indeed, there has been a very limited number of good antibodies for true target detection, such as phosphorylated targets. Therefore, it is difficult to completely capture the entire picture of signal transduction. In addition, the prevalence of mutations per gene is low in general. If the prevalence of mutations is 5% in a population, then it requires at least 100 samples to identify 5 samples with mutations in the gene. In the present study, we compared 12 gene-protein relationships in 42 gene-protein pairs from a 151 gene sequencing panel and 294 RPPA protein panel with a minimum group sample number of three. Although the number of cases was small, the multiregional sampling allowed us to compare the gene-protein relationships in 40-region samples, of which each sample holds gene-protein information. Again, we must admit the comparison is a long shot. However, for instance, we demonstrated that there was an expected result in the present platform for TP53 and p53. Recognizing the limitation from the present study, we are now expanding the number of more than 200 Pan-Cancer sample collection with our university hospital-wide clinical study (MORIOKA study; jRCT1020190017) for network-level gene-protein relationship analyses instead of that of a target protein-level.

6) Does the ctDNA markers perform better than current markers in assessing treat response and disease free survival? Please provide a statistical analysis.

>A long-term prognosis, such as disease-free survival, was not included in the evaluation objects at the time of conceptualization in this observational study. Therefore, except for case presentation, we are still in the process of accumulating longitudinal data of ctDNA monitoring of gastric cancer as a part of our Pan-Cancer project described above (MORIOKA study; jRCT1020190017). In the present study, ctDNA was detected in only three of 10 cases, and therefore we felt that the dataset was too small to compare the clinical validity of ctDNA with currently-used biomarkers, such as CEA. Our separate study of ctDNA monitoring of esophageal squamous cell cancer (ESCC) by Iwaya et al, demonstrated that 91% of ESCC patients benefited from ctDNA monitoring with the clinical validity of an individualized biomarker. The clinical validity was evaluated in terms of early relapse prediction, treatment efficacy evaluation, and non-relapse confirmation, while the clinical validity of conventional serum markers for SCC, CYFRA, and CEA showed much less impact than that of ctDNA, particularly in early relapse prediction (under review; doi: https://doi.org/10.1101/2020.05.01.20087106). In terms of actual survival benefit correlation with ctDNA monitoring, we found that the ESCC group whose ctDNA decreased after treatment had a hazards ratio of 0.1 compared to the group whose ctDNA was sustained. In the current manuscript, we further discuss the clinical utility of ctDNA monitoring in comparison with current biomarkers in terms of treatment response on Page 22, Line 452.

“It seems that not all gastric cancer cases can be monitored by ctDNA alone. Further observational studies are warranted to determine the clinical validity of ctDNA in gastric cancer diagnosis, in comparison with current modalities, such as serum markers and imaging studies.”

Reviewer #2: Good report, will likely significantly advance the use of circulating tumor DNA. Authors are to be commended for careful IRB record notations and excellent supporting online material file, data transparency records.

>Thank you very much for your comment. We are excited to proceed to our next Pan-Cancer study as well as intervention research based on our current data.

---

## [Decision Letter · Decision Letter 1]

17 Sep 2020

Analysis of mutational and proteomic heterogeneity of gastric cancer suggests an effective pipeline to monitor post-treatment tumor burden using circulating tumor DNA

PONE-D-20-11346R1

Dear Dr. Nishizuka,

We’re pleased to inform you that your manuscript has been judged scientifically suitable for publication and will be formally accepted for publication once it meets all outstanding technical requirements.

Kind regards,

Obul Reddy Bandapalli, MSc, PhD

Academic Editor

PLOS ONE

Additional Editor Comments (optional):

Reviewers' comments:

Reviewer's Responses to Questions

**Comments to the Author**

1. If the authors have adequately addressed your comments raised in a previous round of review and you feel that this manuscript is now acceptable for publication, you may indicate that here to bypass the “Comments to the Author” section, enter your conflict of interest statement in the “Confidential to Editor” section, and submit your "Accept" recommendation.

Reviewer #1: All comments have been addressed

Reviewer #2: All comments have been addressed

2. Is the manuscript technically sound, and do the data support the conclusions?

Reviewer #1: Yes

Reviewer #2: Yes

3. Has the statistical analysis been performed appropriately and rigorously? 

Reviewer #1: Yes

Reviewer #2: Yes

4. Have the authors made all data underlying the findings in their manuscript fully available?

Reviewer #1: Yes

Reviewer #2: Yes

5. Is the manuscript presented in an intelligible fashion and written in standard English?

Reviewer #1: Yes

Reviewer #2: Yes

6. Review Comments to the Author

Reviewer #1: (No Response)

Reviewer #2: I had no serious concerns the first time around. I reviewed the authors' responses to other reviewer comments and it seems like the mss, given the changes, is in even better shape.

7. PLOS authors have the option to publish the peer review history of their article (what does this mean?). If published, this will include your full peer review and any attached files.

Reviewer #1: No

Reviewer #2: No

---

## [Editor Report · Acceptance letter]

21 Sep 2020

PONE-D-20-11346R1 

Analysis of mutational and proteomic heterogeneity of gastric cancer suggests an effective pipeline to monitor post-treatment tumor burden using circulating tumor DNA 

Dear Dr. Nishizuka:

I'm pleased to inform you that your manuscript has been deemed suitable for publication in PLOS ONE. Congratulations! Your manuscript is now with our production department. 

Kind regards, 

on behalf of

Dr. Obul Reddy Bandapalli 

Academic Editor

PLOS ONE